# Efficacy of sanitization protocols in removing parasites in vegetables: A protocol for a systematic review with meta-analysis

Cláudio Márcio de Medeiros Maia[1], Karla Suzanne Florentino da Silva Chaves Damasceno[1], Larissa Mont'Alverne Jucá Seabra[1], Gabriela Chaves[2], Lívia Maria da Costa Dantas[3], Francisco Canindé de Sousa Júnior[1,3], Cristiane Fernandes de Assis[1,3]*

1 Nutrition Postgraduate Program, Center for Health Sciences, Federal University of Rio Grande do Norte, Natal, Brazil, 2 Myant INC, Research and Development, Toronto, Canada, 3 Department of Pharmacy, Center for Health Sciences, Federal University of Rio Grande do Norte, Natal, Brazil

* crisstianeassis@hotmail.com

## Abstract

### Background

Parasitic contamination of vegetables is a public health problem in several countries and a challenge for food safety. With a short path from the field to the table, these foods can suffer several flaws in the good practices of production, transport and packaging which culminate in an offer of contaminated food to consumers. Therefore, this study describes a systematic review protocol with meta-analysis on evaluating the effectiveness of existing sanitation methods in removing parasites from vegetables.

### Methods

The study will be conducted from published studies that report analyzes of parasites in vegetables before and after sanitization processes. The MEDLINE, Embase, Web of Science, FSTA, LILACS, Scopus and AGRIS electronic databases will be used. In addition, manual searches will be carried out through related articles, references to included articles and directories of theses and dissertations. The primary outcome will be the reduction or absence of parasitic forms in vegetables after the intervention or combined interventions, and the secondary outcomes will include: identification of the main parasites, assessment of the time required for processing and cost-effectiveness analysis. Two authors will independently screen the studies and extract data. Disagreements will be resolved by discussion, and a third reviewer will decide if there is no consensus. The criteria established by the Cochrane Manual (with some adaptations) will be used to assess the risk of bias in the studies and if the results are considered acceptable and sufficiently homogeneous, and a meta-analysis will be performed to synthesize the findings.

### Discussion

The systematic review produced from this protocol will provide evidence on the effectiveness of sanitation protocols for removing parasitic forms in vegetables and will contribute to

**Funding:** Funding:This work received financial support from the Coordenação de Aperfeiçoamento de Pessoal de Nível Superior,

which granted the scholarship (001). The funders had no role in study design, data collection and analysis, decision to publish, or preparation of the manuscript.

**Competing interests:** The authors have declared that no competing interests exist.

strengthening food safety, with the adoption of best sanitation practices and prevention of health risks. In addition, the study may highlight possible knowledge gaps that need to be filled with new research.

## Systematic review registration

PROSPERO registration number: CRD42020206929.

## Introduction

There are multiple ecological and environmental factors in the vegetable production stages. They range from soil contamination with animal and human waste; poor irrigation conditions and prolonged periods of rain, which carry parasites to contaminate underground soil; low quality of fertilizer and chemical components; and direct contact with animals and insects, increasing the diffusion process of enteroparasitosis [1].

In addition to these factors, associated with the increased consumption of raw or under-cooked vegetables in recent years, expansion of the volume in world trade, and also the persistence of protozoa in contaminated vegetables further increase the risk of contracting foodborne diseases and reflect the potential that parasitic agents present as a public health problem [2].

Although the stage of vegetable cultivation is probably the main source of contamination, other factors influence the contamination of vegetables during the production chain: improper handling under low hygienic conditions, low quality of water used in post-harvest cleaning, transportation, contaminated equipment and temperature abuse in storage, product display at the point of sale and even handling in the home kitchen are stages sensitive to contamination and require intensification of health surveillance strategies [3–5].

Even with these risks throughout the entire production chain, the correct hygiene of these foods can reduce or completely eliminate contamination levels [6]. There are several methods available to carry out this decontamination: physical methods (brushing, rinsing, irradiation), chemical methods (sodium hypochlorite, acidified sodium chlorite, acids, hydrogen peroxide, chlorine dioxide, bromine, iodine, trisodium phosphate, compounds of quaternary ammonium and ozone) and even biological methods when using antagonists as biocontrol agents to fight a specific pathogen [7].

The procedure choice should be based on the absence of damage to plant structures and safety to handlers (being non-toxic, non-irritating and non-corrosive), presenting good solubility for residue removal, good storage stability, in addition to being economical and having quick action [7, 8].

There is no consensus on the procedures required to sanitize vegetables. The general guidelines on food hygiene provided by the *Codex Alimentarius* establish for vegetable sanitization, in addition to cleaning in water to remove dirt, the use of detergent solutions and appropriate chemical sanitizers, which must have the correct removal of residues after sanitization [9]. The FDA and the CDC recommend that vegetables should be washed in running water, avoiding the use of chemicals products like soap, detergent and any use of bleach solution, and for products labeled as ready-to-eat, this cleaning in running water can be dispensed [10, 11]. In Brazil, the protocol for sanitizing vegetables recommended by the Ministry of Health of Brazil recommends washing vegetables in running water, with subsequent immersion of these foods in 200ppm chlorinated water for 15–30 minutes and removal of residues in running water [12].

These protocols, although efficient in inactivating or decreasing the bacterial load [13–15], have questionable efficacy in parasitological decontamination [16–20]. One of the factors that increase the difficulty of this decontamination is due to the surface characteristics of the vegetables. Green leafy vegetables with a wide and uneven surface promote the attachment of parasite eggs and cysts that easily adhere to the matrix of these foods, in contrast to vegetables with smooth and narrow surfaces with lower parasite rates due to the reduction of this attachment of parasite forms [21]. One example is the high adherence of *Cryptosporidium* oocysts to spinach, with the internalization of the oocysts in the leaf stomata, making removal by washing alone inefficient [22].

This questionable effectiveness of the various methods available for removing parasites from vegetables makes a systematic review on the subject necessary. Therefore, the aim of this study is to describe a protocol for conducting a systematic review with meta-analysis which will assess the published evidence about the best hygiene practices that reduce parasitic contamination in vegetables, ensuring better food quality.

## Materials and methods

### Study design

This protocol was registered in the PROSPERO database (International prospective register of systematic reviews), with registration number CRD42020206929 and is reported in accordance with the guidelines provided in the PRISMA-P guide (Preferred Reporting Items for Systematic Review and Meta-Analyses Protocol) [23] (S1 Checklist). A systematic review protocol should present the methodological strategies that will be used in the systematic review, such as: search strategies, eligibility criteria, what data will be extracted from the selected articles, what are the variables of interest, how the data will be analyzed, and how heterogeneities will be handled, thus, the protocol should demonstrate transparency in the process of performing the systematic review [24].

The systematic review with meta-analysis will be conducted following the recommendations of the Cochrane Collaboration Handbook of Systematic Reviews [25], as a systematic, transparent and reproducible method in the investigation of scientific evidence, following the steps corresponding to the flowchart below (Fig 1).

### Eligibility criteria

The eligibility criteria for the study were defined using the PICOS classification (Population, Intervention, Comparison, Outcome, Study design) as a tool to guide the research and formulate the search strategy, as described in Table 1. No linguistic or date restriction were applied as part of the eligibility criteria.

### Search strategy

Structured terms were created based on information from PICOS that would translate the search criteria into the formulation of a search strategy. The bibliographic databases chosen were: MEDLINE, Embase, Web of Science, FSTA (Food Science and Technology Abstracts), LILACS (Latin American and Caribbean Health Sciences), Scopus, AGRIS (International Information System for Agricultural Sciences and Technology), in addition to manual searches in related articles and thesis and dissertation directory.

The primary search strategy was: ("Lettuce" OR "Vegetables" OR "Fresh produce" OR "Plants, Edible" OR "Leafy vegetables" OR "Salads"), **AND** ("exp Disinfection/" OR "hypochlorite.mp." OR "Hypochlorous Acid/" OR "Peracetic Acid/" OR "Hydrogen Peroxide/" OR

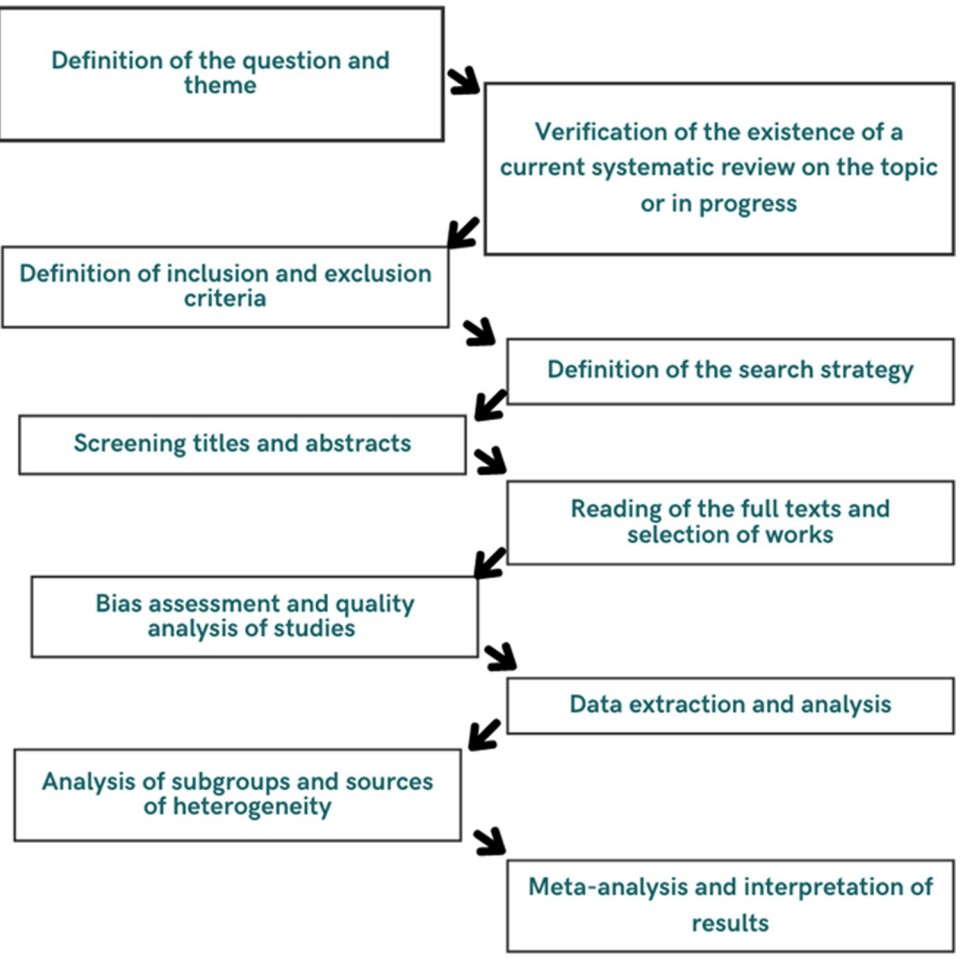

**Fig 1. Flowchart steps of the systematic review.**

"exp Detergents/" OR "Chlorine/" OR "disinfection agent.mp." OR "Decontamination/" OR "chemical agent.mp." OR "Sanitation/" OR "Wash" OR "Clean*.mp" OR "Anti-Infective Agents/" OR "Ozone" OR "Ultraviolet "OR "Disinfectants"), **AND** ("Parasites" OR "Parasite Load" OR "Parasit*.mp." OR "Parasite Encystment" OR "exp Oocysts/" OR "exp Helminths/" OR "Egg Count" OR "Parasite Egg Count/" OR "parasite examination.mp." OR "intestine parasite.mp." OR "Entamoeba.mp. Dysentery" OR "Amebic Entamoebiasis" OR "Food safety" OR "Food quality" OR "Food analysis" OR "Food contamination" OR "Food parasitology").

**Table 1. Eligibility criteria.**

| Category | Inclusion | Exclusion |
|---|---|---|
| Population | Studies that report parasitic analyzes on vegetables before and after some intervention process to sanitize these foods and compare this effect. | Studies on the prevalence of parasites in vegetables. |
| Intervention | Action of different sanitization protocols (chemical, physical or associated) in removing or reducing parasitic forms adhered to vegetables. | Studies which only report the sanitizing effect on microbiological contamination of vegetables. |
| Comparator | Unsanitized vegetables or samples with artificial contamination. | |
| Outcomes | Absence or reduction of parasitic forms in vegetables. | |
| Study type | Comparative and controlled studies | |

The primary search strategy had some modifications to meet the specifics in the search syntax of each database (S1 File).

## Screening procedure

The screening procedure will be carried out in two stages and by two evaluators independently. The titles and abstracts will be read in the first stage, and the full texts will be read to confirm eligibility if the study meets the inclusion criteria; a third reviewer will decide on the inclusion of the article in case of disagreement. The Mendeley software program will be used to manage references to carry out this selection and remove duplicate articles. The screening will be done without any interference or contact between reviewers in order to maintain transparency and avoid influences on the decision process.

## Data extraction

Data extraction will be performed by filling out an electronic spreadsheet with a detailed description of the main information of the selected studies and will also be performed by two researchers independently in order to avoid the measurement bias, which occurs due to misinterpreting information or even the loss of important data collection. Authors of articles may be contacted for clarification if information is insufficient. Any cases of disagreement will also be resolved by consensus, and when disagreement persists, the definition will be given by a third evaluator.

The data to be collected in order for this information to be clearly and objectively compared was previously established, thus facilitating the agreement and disagreement process, including: quantity and types of vegetables analyzed, type of intervention used, description of the methodologies used in the phases of extraction and identification, which parasitic forms were found, the outcomes obtained, in addition to the time and materials needed for processing.

## Risk of bias assessment

The qualitative analysis of the articles will be performed by assessing the risk of bias, based on the assessment methodology of the Cochrane Risk of Bias (RoB) 2.0 Tool [26], but with some modifications to suit food analysis studies such as verifying the sensitivity and specificity of the analytical method; using the test according to validated methodology or if it contains validated modifications; number of samples analyzed and verification of randomization, blinding or random numbering of the sample that minimizes the influence of the analyst.

The evaluation will be categorized into 5 domains: selection bias, sampling bias, performance bias, detection bias and reporting bias. A pilot analysis will initially be carried out among the evaluators to ensure that they can apply the criteria consistently. Again, this step will be performed independently by two evaluators, and disagreements between researchers will be decided by a third evaluator.

Each evaluator's judgment in this study will be scored according to the criteria established in the Bias Assessment Worksheet (S2 File), which comprises a judgment and support for each bias domain. The judgment of each domain involves assessing the risk of bias as "low", "high" or "uncertain", with the latter occurring when the information is not possible for judgment in other categories or when the bias is assessed as an intermediary in its magnitude power.

The points will be added up for the overall evaluation of the article and ranked according to the following score: considered "low risk of bias" if the score is between 7 to 10 points; considered "uncertain risk of bias" if the score is between 3 to 6 points; and considered "high risk of bias" if it is below 2 points. The level of bias will influence the importance degree of the study in the evidence synthesis from this systematic review.

## Data synthesis

The main outcome to be evaluated will be the effectiveness of sanitation protocols in parasitic decontamination, which will be calculated by the percentage of parasitic forms that were reduced or completely eliminated after the sanitization process. In addition, secondary outcomes will also be analyzed such as an identification of the main parasites found, evaluation of the time required for processing and cost-effectiveness analysis. Thus, the average price of inputs needed for vegetable hygiene will be calculated for this analysis, and then compared with the amount of larvae, eggs or parasite cysts that were effectively reduced after processing.

It is expected that the measurement effects are quite different as this is such a variable population group, even due to the characteristics of the foods, as in the case of leafy vegetables, which have a larger contact surface for fixing parasites than other vegetables. Therefore, analyses will be performed in subgroups if possible to minimize this problem, reducing the effect of heterogeneity between the results. This analysis will depend on how the outcomes will be made available in the studies and on the amount of work available.

Other possible points of heterogeneity in the research such as: study design, types of control, analytical methodologies, validation of results and statistical analysis will also be treated in subgroups when possible.

Heterogeneity will be assessed using i-squared ($I^2$) and chi-squared ($\chi^2$) statistics, which describe whether the percentage of variation between studies is due to heterogeneity and not to chance. The $\chi^2$ test is one of the most used to assess the significance level of heterogeneity, if there is little variation between the tests, with a significance level of $p < 0.10$ and values of the magnitude of heterogeneity, assessed by $I^2$, less than 50%, we can conclude that the synthesis of results is viable [27, 28]. Sensitivity analysis will be performed considering the studies that present high bias risk versus low bias risk for the domains (sampling, performance and detection bias).

The potential for publication bias will be analyzed using Funnel plots by graphical verification of asymmetry between the included studies. In addition, additional statistical tests by Egger [29] and Begg [30] will be performed, which will consider the presence of publication bias for p-values $< 0.10$.

In the systematic review, the data regarding the efficacy rates in reducing parasitic forms in vegetables of each sanitization protocol will be used in the comparison between the protocols, and if the results are considered acceptable and sufficiently homogeneous, we will use the Review Manager software program (RevMan version 5.4) for meta-analysis. For the meta-analysis result, the data will be presented estimating a 95% confidence level (95% CI) for corresponding effect size.

If quantitative synthesis is not appropriate, the results will be summarized and discussed, weighing the risk of bias and the magnitude of the findings of each study. After synthesizing the results, interventions which present effective results and provide possible conclusions for conducting further studies and better decision-making will be identified.

## Discussion

Infections transmitted by parasites in food have a global distribution, and although they are preventable, they cause significant morbidities, ranging from mild to severe, and in some cases resulting in mortality. In addition, there is the associated deleterious effect on the socio-economy, worker absence due to illness, lowered productivity in subclinical infections, and the costs of treatments [31]. One of the factors that contributes to their transmission is the neglect to combat the disease effectively, perhaps due to the low presence of acute symptoms, which

affects the delay in treatment and expands its spread, as well as the mistaken perception that they are diseases which are only related to poverty [32].

In addition, the increase in the global vegetable trade and changes in eating habits with the increase in meals away from home or delivered by apps, in addition to the demand for ready-to-eat foods such as peeled and portioned vegetables, further favor the transmission risk of these diseases [3].

Therefore, carrying out studies that assess deficiencies in food hygiene practices by agricultural producers, vendors and consumers and which point to improvements in the process and in the education of the actors involved are very important, as they favor an increase in the general perception of risks to food safety and facilitate the adoption of best practices, ensuring improvement in food quality [4].

One of the main advantages of conducting a systematic review protocol is to enhance the commitment to transparency in research by publicizing the methodological construction of the systematic review in a broad way, structured, and organized enough to provide robust findings that aid in the decision making process and that are strongly valued by all stakeholders. Therefore, researching the effectiveness of sanitization protocols through a systematic review will serve to provide sufficient scientific evidence to contribute to the prevention of parasitic diseases obtained by these foods, as well as an alternative in combating the prevalence of enteroparasites in vegetables.

One of the major limiting factors of this protocol is the restriction to unpublished, ongoing, gray literature, or unregistered data sources in the chosen databases, which thus will not be included in the synthesis of results. Evidently, methodological limitations of primary studies such as the use of manual and insensitive diagnostic methods, reporting biases, and methodological modifications without prior validation may provide results with lower confidence and should have the quality of their evidence evaluated to minimize these effects.

The expectation is that this protocol will assist in carrying out a systematic review, ensuring greater reliability in the results through transparency in the criteria and methodology adopted, as well as facilitating reproducibility of future updates of this review, contributing to disseminate more systematic reviews in the area of controlling food quality.

## Supporting information

**S1 Checklist. PRISMA-P 2015 checklist.**
(DOCX)

**S1 File.**
(DOCX)

**S2 File. Risk of bias evaluation.**
(DOCX)

## Author Contributions

**Conceptualization:** Gabriela Chaves.

**Data curation:** Gabriela Chaves.

**Formal analysis:** Cláudio Márcio de Medeiros Maia.

**Investigation:** Lívia Maria da Costa Dantas.

**Methodology:** Lívia Maria da Costa Dantas.

**Project administration:** Karla Suzanne Florentino da Silva Chaves Damasceno, Cristiane Fernandes de Assis.

**Resources:** Cláudio Márcio de Medeiros Maia, Lívia Maria da Costa Dantas.

**Supervision:** Karla Suzanne Florentino da Silva Chaves Damasceno, Larissa Mont'Alverne Jucá Seabra, Cristiane Fernandes de Assis.

**Validation:** Gabriela Chaves.

**Visualization:** Larissa Mont'Alverne Jucá Seabra, Francisco Canindé de Sousa Júnior.

**Writing – original draft:** Cláudio Márcio de Medeiros Maia, Cristiane Fernandes de Assis.

**Writing – review & editing:** Karla Suzanne Florentino da Silva Chaves Damasceno, Larissa Mont'Alverne Jucá Seabra, Gabriela Chaves, Francisco Canindé de Sousa Júnior.

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
