## [Decision Letter · Decision Letter 0]

3 Feb 2022

PONE-D-21-34531Efficacy of sanitization protocols in removing parasites in vegetables: A protocol for a systematic review with meta-analysisPLOS ONE

Dear Dr. Cristiane Fernandes de Assis,

Thank you for submitting your manuscript to PLOS ONE. After careful consideration, we feel that it has merit but does not fully meet PLOS ONE’s publication criteria as it currently stands. Therefore, we invite you to submit a revised version of the manuscript that addresses the points raised during the review process.

ACADEMIC EDITOR:

Dear Dr. Cristiane Fernandes de Assis,

Based on the advice received, I feel that your manuscript could be reconsidered for publication should you be prepared to incorporate changes as suggested by reviewers. When preparing your revised manuscript, you are asked to carefully consider the reviewer comments which can be found below, and submit a list of responses to the comments. The final decision will be taken after your response letter and revision.

We look forward to receiving your revised manuscript.

Kind regards,

Masoud Foroutan, Ph.D; Assistant Professor

Academic Editor

PLOS ONE

 “This work received financial support from the Coordenação de Aperfeiçoamento de Pessoal de Nível Superior, which granted the scholarship (001).”

Reviewers' comments:

Reviewer's Responses to Questions

**Comments to the Author**

1. Does the manuscript provide a valid rationale for the proposed study, with clearly identified and justified research questions?

Reviewer #1: Yes

Reviewer #2: Yes

Reviewer #3: Yes

Reviewer #4: No

2. Is the protocol technically sound and planned in a manner that will lead to a meaningful outcome and allow testing the stated hypotheses?

Reviewer #1: Yes

Reviewer #2: Yes

Reviewer #3: No

Reviewer #4: No

3. Is the methodology feasible and described in sufficient detail to allow the work to be replicable?

Reviewer #1: Yes

Reviewer #2: Yes

Reviewer #3: No

Reviewer #4: No

4. Have the authors described where all data underlying the findings will be made available when the study is complete?

Reviewer #1: Yes

Reviewer #2: Yes

Reviewer #3: No

Reviewer #4: No

5. Is the manuscript presented in an intelligible fashion and written in standard English?

Reviewer #1: Yes

Reviewer #2: Yes

Reviewer #3: Yes

Reviewer #4: No

6. Review Comments to the Author

You may also provide optional suggestions and comments to authors that they might find helpful in planning their study.

Reviewer #1: This paper describes a protocol for a systematic review of published evidence on best hygiene practices to reduce parasite contamination of vegetables through a meta-analysis. This has certain significance for food quality assurance.

Specific comment

Line 52-91: The references in this part are too old. It is suggested to cite more literatures published in the recent 2-3 years.

Line 109-112: The first column of Table 1 has five pieces of content. Why are they classified in this way? It is suggested to introduce the basis for such classification.

Line 120-131: Keywords within the five categories are linked with “OR” during search. However, it is not clear whether to use “OR” or “AND” to connect the relationship between categories.

Line 171: “Kahren.2023” and “Kahren.2024” appear in the first table of supplementary material 2. What are the meanings of “2023” and “2024”?

Line 204-205: Sensitivity analysis is recommended to ensure the robustness of the results obtained under certain conditions.

Line 218-240: What are the advantages and limitations of this protocol? In short, the discussion section is poor.

Reviewer #2: In your data synthesis you have stated that : ''the main outcome to be evaluated will be the effectiveness of sanitation protocols in parasitic decontamination, which will be calculated by the percentage of parasitic forms that were reduced or completely eliminated after the sanitization process......'' My concern is that :different studies might have used different laboratory techniques to check the presence /absence of parasite stages on vegetables after the produce is went through sanitization protocol ...so how are you going to manage such issues ?

Reviewer #3: In this manuscript, the authors highlight the Efficacy of sanitization protocols in removing parasites in vegetables: A protocol for a systematic review with meta-analysis. Overall, the topic is interesting and important; however, the manuscript has major drawbacks and limitations related to data management and statistical analysis, methods, presentation of results and quality of writing.

I have some considerations.

1. I do not see any PRISMA flow diagram in the text. Since the authors searched other databases e.g. WHO and performed other manual searches, they ought to use the following PRISMA flow diagram: PRISMA 2020 flow diagram for new systematic reviews, which include searches of databases, registers and other sources. Manual searching can be conducted by tracking new citations, using keywords in the databases, and even tracing the sources of news I have added the flow diagram at the bottom. Please look at this paper as a reference for step-by-step guidelines on how to conduct a systematic review and meta-analysis: (https://pubmed.ncbi.nlm.nih.gov/31388330/)

2. I could not find any Forest plots, Egger’s funnel plot and Begg’s funnel plot in the text. Please provide these items.

3. All scientific names in the manuscript must be in italic.

4. Please update your information based on the two new review articles (https://doi.org/10.1016/j.foodcont.2021.108582 & https://doi.org/10.1016/j.foodcont.2021.108656)

5. Why the authors of the manuscript did not include the start year of study?

6. Provide the syntax for each databases appropriately.

7. Please consider upper and lower limits e.g. 0% (95% CI; 0% - 1%) this throughout the manuscript (Abstract, Result, Discussion,…), and supplementary files.

8. The discussion section of the article is very short, please extend this section.

9. Limitations of the study should be acknowledged and discussed at the end of the discussion section.

10. Provide author contributions statement after acknowledgments.

11. Finally, all the analyzed studies must be in the references chapter.

Reviewer #4: PLOS ONE

Efficacy of sanitization protocols in removing parasites in vegetables: A protocol for a

systematic review with meta-analysis

--Manuscript Draft

REVIEW

The idea to prepare this metanalysis is excellent.

However, looking in the obtained file I can see any results and data analysis.

The text in the MS needs to be revised. The authors speak in the future despite they need to speak in the past.

I cant see any results….

I think that the protocol should be also compared with the protocol developed in Europe and in the USA

Where is the aim of the study promised in the lines 86-91?

I can not open any supplementary material.

7. PLOS authors have the option to publish the peer review history of their article (what does this mean?). If published, this will include your full peer review and any attached files.

Reviewer #1: No

Reviewer #2: No

Reviewer #4: No

---

## [Author Response · Author response to Decision Letter 0]

18 Mar 2022

Dear Editor,

 Thank you for considering our manuscript entitled "Efficacy of sanitization protocols in removing parasites in vegetables: A protocol for a systematic review with meta-analysis", for revision. We have carefully read the comments, answering all questions in this letter, and made the requested changes in the revised manuscript (highlighted in yellow). We hope the revised manuscript is now suitable for publication in PLos One. 

Best regards,

Cristiane Fernandes de Assis 

Department of Pharmacy 

Federal University of Rio Grande do Norte

Phone: (55 84) 99152-7007

E-mail: cristianeassis@hotmail.com

Answers to reviewer #1: This paper describes a protocol for a systematic review of published evidence on best hygiene practices to reduce parasite contamination of vegetables through a meta-analysis. This has certain significance for food quality assurance.

Specifics comments

Line 52-91: The references in this part are too old. It is suggested to cite more literatures published in the recent 2-3 years.

References have been updated.

Line 109-112: The first column of Table 1 has five pieces of content. Why are they classified in this way? It is suggested to introduce the basis for such classification.

This division into 5 parts was established according to the anagram PICOS, as mentioned in line 124.

Line 120-131: Keywords within the five categories are linked with “OR” during search. However, it is not clear whether to use “OR” or “AND” to connect the relationship between categories.

Thank you for the comment. The suggestions were accepted. 

Line 171: “Kahren.2023” and “Kahren.2024” appear in the first table of supplementary material 2. What are the meanings of “2023” and “2024”?

Thank you for your comment. The file has been replaced, the mentioned information refers to examples of bias evaluation that were sent in the submission by mistake. The table for bias evaluation analysis is now described in supplementary material 3. We apologize for that.

Line 204-205: Sensitivity analysis is recommended to ensure the robustness of the results obtained under certain conditions.

A topic about sensitivity analysis was included in the data synthesis section (page 09).

Line 218-240: What are the advantages and limitations of this protocol? In short, the discussion section is poor.

Thank you for the comment. The suggestions were accepted. We made the proper changes.

Answers to reviewer #2: 

In your data synthesis you have stated that : ''the main outcome to be evaluated will be the effectiveness of sanitation protocols in parasitic decontamination, which will be calculated by the percentage of parasitic forms that were reduced or completely eliminated after the sanitization process......'' My concern is that :different studies might have used different laboratory techniques to check the presence /absence of parasite stages on vegetables after the produce is went through sanitization protocol ...so how are you going to manage such issues ?

In the last paragraph of the data synthesis (lines 235 - 238) it explains how the data are treated if the quantitative synthesis is not appropriate due to the reporting of results with too much heterogeneity. These results will be summarized and discussed, considering the risk of bias and the magnitude of the findings of each study.

Answers to reviewer #3: In this manuscript, the authors highlight the Efficacy of sanitization protocols in removing parasites in vegetables: A protocol for a systematic review with meta-analysis. Overall, the topic is interesting and important; however, the manuscript has major drawbacks and limitations related to data management and statistical analysis, methods, presentation of results and quality of writing.I have some considerations.

1. I do not see any PRISMA flow diagram in the text. Since the authors searched other databases e.g. WHO and performed other manual searches, they ought to use the following PRISMA flow diagram: PRISMA 2020 flow diagram for new systematic reviews, which include searches of databases, registers and other sources. Manual searching can be conducted by tracking new citations, using keywords in the databases, and even tracing the sources of news I have added the flow diagram at the bottom. Please look at this paper as a reference for step-by-step guidelines on how to conduct a systematic review and meta-analysis: (https://pubmed.ncbi.nlm.nih.gov/31388330/)

Dear reviewer, this manuscript is a protocol for a systematic review, it is not yet the systematic review, it describes the methodology that will be applied in the systematic review. We follow the Prisma-P checklist (PRISMA for systematic review protocols ), the most current one published in 2015. (http://www.prisma-statement.org/Extensions/Protocols).

2. I could not find any Forest plots, Egger’s funnel plot and Begg’s funnel plot in the text. Please provide these items.

Dear reviewer, as this manuscript is a protocol for a systematic review, therefore, Forest plots, Egger's funnel plot and Begg's funnel plot will be placed in the final systematic review if the results have a compatible heterogeneity.

3. All scientific names in the manuscript must be in italic.

Thank you for the comment. The suggestions were accepted. 

4. Please update your information based on the two new review articles (https://doi.org/10.1016/j.foodcont.2021.108582 & https://doi.org/10.1016/j.foodcont.2021.108656)

5. Why the authors of the manuscript did not include the start year of study?

Because few studies were selected according to the inclusion criteria, we decided to expand the sensitivity of the search by not restricting publication dates.

6. Provide the syntax for each databases appropriately.

The syntaxes have been included in supplementary material 2.

7. Please consider upper and lower limits e.g. 0% (95% CI; 0% - 1%) this throughout the manuscript (Abstract, Result, Discussion,…), and supplementary files.

Thank you for the comment. The suggestions were accepted.

8. The discussion section of the article is very short, please extend this section.

Thank you for the comment. The suggestions were accepted.

9. Limitations of the study should be acknowledged and discussed at the end of the discussion section.

Thank you for the comment. The suggestions were accepted.

10. Provide author contributions statement after acknowledgments.

Thank you for the comment. The suggestions were accepted.

11. Finally, all the analyzed studies must be in the references chapter.

The results will be presented in the publication of the systematic review, as mentioned this manuscript is the protocol for the systematic review.

Answers to reviewer #4:

The idea to prepare this metanalysis is excellent. However, looking in the obtained file I can see any results and data analysis. The text in the MS needs to be revised. The authors speak in the future despite they need to speak in the past. I cant see any results….I think that the protocol should be also compared with the protocol developed in Europe and in the USA Where is the aim of the study promised in the lines 86-91? I can not open any supplementary material.

 Dear reviewer, this manuscript is a protocol, not the systematic review yet, hence, the absence of analysis, results and future tense in the text.

---

## [Decision Letter · Decision Letter 1]

14 Apr 2022

PONE-D-21-34531R1Efficacy of sanitization protocols in removing parasites in vegetables: A protocol for a systematic review with meta-analysisPLOS ONE

Dear Dr. Fernandes de Assis,

Thank you for submitting your manuscript to PLOS ONE. After careful consideration, we feel that it has merit but does not fully meet PLOS ONE’s publication criteria as it currently stands. Therefore, we invite you to submit a revised version of the manuscript that addresses the points raised during the review process.

ACADEMIC EDITOR:

We look forward to receiving your revised manuscript.

Kind regards,

Masoud Foroutan, Ph.D; Assistant Professor

Academic Editor

PLOS ONE

Journal Requirements:

Reviewers' comments:

Reviewer's Responses to Questions

**Comments to the Author**

1. Does the manuscript provide a valid rationale for the proposed study, with clearly identified and justified research questions?

Reviewer #1: Yes

Reviewer #2: Yes

Reviewer #3: Yes

Reviewer #4: Yes

2. Is the protocol technically sound and planned in a manner that will lead to a meaningful outcome and allow testing the stated hypotheses?

Reviewer #1: Yes

Reviewer #2: Yes

Reviewer #3: Yes

Reviewer #4: Yes

3. Is the methodology feasible and described in sufficient detail to allow the work to be replicable?

Reviewer #1: Yes

Reviewer #2: Yes

Reviewer #3: Yes

Reviewer #4: Yes

4. Have the authors described where all data underlying the findings will be made available when the study is complete?

Reviewer #1: Yes

Reviewer #2: Yes

Reviewer #3: Yes

Reviewer #4: Yes

5. Is the manuscript presented in an intelligible fashion and written in standard English?

Reviewer #1: Yes

Reviewer #2: Yes

Reviewer #3: Yes

Reviewer #4: Yes

6. Review Comments to the Author

You may also provide optional suggestions and comments to authors that they might find helpful in planning their study.

Reviewer #1: Present study describes a systematic review protocol with meta-analysis on evaluating the effectiveness of existing sanitation methods in removing parasites from vegetables it will help researcher to better understanding impotant of food safety. The authors already answered my questions, and revised them in the text. recommend it to publishing.

Reviewer #2: Generally , I want to mention again that the topic is interesting and important. The authors have made major modifications by taking into consideration reviewers comment .Specifically , my concern about the protocol is well addressed.

Reviewer #3: The authors have answered all of my questions, and made considerable revisions to the manuscript and can be accepted.

Reviewer #4: The authors improved their MS.

Some positions in the refs need revision; see 21, 31,

pl also check the international literature if you are missing some original related papers.

7. PLOS authors have the option to publish the peer review history of their article (what does this mean?). If published, this will include your full peer review and any attached files.

Reviewer #1: No

Reviewer #2: No

Reviewer #3: No

Reviewer #4: No

---

## [Author Response · Author response to Decision Letter 1]

22 Apr 2022

Federal University of Rio Grande do Norte 

Departament of Pharmacy 

Natal, April 22th, 2022

Dear Editor,

 Thank you for considering our manuscript entitled "Efficacy of sanitization protocols in removing parasites in vegetables: A protocol for a systematic review with meta-analysis" for revision. We have made the changes suggested by the reviewers (highlighted in yellow). We hope the revised manuscript is now suitable for publication in PLoS One. 

Best regards,

Cristiane Fernandes de Assis 

Department of Pharmacy 

Federal University of Rio Grande do Norte

Phone: (55 84) 99152-7007

E-mail: cristianeassis@hotmail.com

Answers to reviewer #1: Present study describes a systematic review protocol with meta-analysis on evaluating the effectiveness of existing sanitation methods in removing parasites from vegetables it will help researcher to better understanding important of food safety. The authors already answered my questions and revised them in the text. recommend it to publishing.

Thank you for considering the article for publication.

Answers to reviewer #2: Generally, I want to mention again that the topic is interesting and important. The authors have made major modifications by taking into consideration reviewers comment. Specifically, my concern about the protocol is well addressed.

Thank you for considering the article for publication.

Answers to reviewer #3: The authors have answered all of my questions and made considerable revisions to the manuscript and can be accepted.

Thank you for considering the article for publication.

Answers to reviewer #4: The authors improved their MS. Some positions in the refs need revision; see 21, 31, pl also check the international literature if you are missing some original related papers.

We appreciate the comments and made the proper changes in references.

---

## [Editor Report · Decision Letter 2]

26 Apr 2022

Efficacy of sanitization protocols in removing parasites in vegetables: A protocol for a systematic review with meta-analysis

PONE-D-21-34531R2

Dear Dr. de Assis,

We’re pleased to inform you that your manuscript has been judged scientifically suitable for publication and will be formally accepted for publication once it meets all outstanding technical requirements.

Kind regards,

Masoud Foroutan, Ph.D; Assistant Professor

Academic Editor

PLOS ONE

---

## [Editor Report · Acceptance letter]

28 Apr 2022

PONE-D-21-34531R2 

Efficacy of sanitization protocols in removing parasites in vegetables: A protocol for a systematic review with meta-analysis 

Dear Dr. de Assis:

I'm pleased to inform you that your manuscript has been deemed suitable for publication in PLOS ONE. Congratulations! Your manuscript is now with our production department. 

Kind regards, 

on behalf of

Dr. Masoud Foroutan 

Academic Editor

PLOS ONE